# Gaussian Attention Model and Its Application to Knowledge Base Embedding and Question Answering

**Liwen Zhang**
Department of Computer Science
University of Chicago
Chicago, IL 60637, USA
liwenz@cs.uchicago.edu

**John Winn & Ryota Tomioka**
Microsoft Research Cambridge
Cambridge, CB1 2FB, UK
{jwinn, ryoto}@microsoft.com

## Abstract

We propose the Gaussian attention model for content-based neural memory access. With the proposed attention model, a neural network has the additional degree of freedom to control the focus of its attention from a laser sharp attention to a broad attention. It is applicable whenever we can assume that the distance in the latent space reflects some notion of semantics. We use the proposed attention model as a scoring function for the embedding of a knowledge base into a continuous vector space and then train a model that performs question answering about the entities in the knowledge base. The proposed attention model can handle both the propagation of uncertainty when following a series of relations and also the conjunction of conditions in a natural way. On a dataset of soccer players who participated in the FIFA World Cup 2014, we demonstrate that our model can handle both path queries and conjunctive queries well.

## 1 Introduction

There is a growing interest in incorporating external memory into neural networks. For example, memory networks (Weston et al., 2014; Sukhbaatar et al., 2015) are equipped with static memory slots that are content or location addressable. Neural Turing machines (Graves et al., 2014) implement memory slots that can be read and written as in Turing machines (Turing, 1938) but through differentiable attention mechanism.

Each memory slot in these models stores a vector corresponding to a continuous representation of the memory content. In order to recall a piece of information stored in memory, attention is typically employed. Attention mechanism introduced by Bahdanau et al. (2014) uses a network that outputs a discrete probability mass over memory items. A memory read can be implemented as a weighted sum of the memory vectors in which the weights are given by the attention network. Reading out a single item can be realized as a special case in which the output of the attention network is peaked at the desired item. The attention network may depend on the current context as well as the memory item itself. The attention model is called location-based and content-based, if it depends on the location in the memory and the stored memory vector, respectively.

Knowledge bases, such as WordNet and Freebase, can also be stored in memory either through an explicit knowledge base embedding (Bordes et al., 2011; Nickel et al., 2011; Socher et al., 2013) or through a feedforward network (Bordes et al., 2015).

When we embed entities from a knowledge base in a continuous vector space, if the capacity of the embedding model is appropriately controlled, we expect semantically similar entities to be close to each other, which will allow the model to generalize to unseen facts. However the notion of proximity may strongly depend on the type of a relation. For example, Benjamin Franklin was an engineer but also a politician. We would need different metrics to capture his proximity to other engineers and politicians of his time.

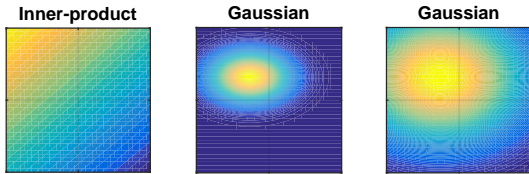

Figure 1: Comparison of the conventional content-based attention model using inner product and the proposed Gaussian attention model with the same mean but two different covariances.

In this paper, we propose a new attention model for content-based addressing. Our model scores each item $\boldsymbol{v}_{\texttt{item}}$ in the memory by the (logarithm of) multivariate Gaussian likelihood as follows:

$$
\begin{aligned}
\text{score}(\boldsymbol{v}_{\texttt{item}}) &= \log \phi(\boldsymbol{v}_{\texttt{item}} | \boldsymbol{\mu}_{\texttt{context}}, \boldsymbol{\Sigma}_{\texttt{context}}) \\
&= -\frac{1}{2}(\boldsymbol{v}_{\texttt{item}} - \boldsymbol{\mu}_{\texttt{context}}) \boldsymbol{\Sigma}_{\texttt{context}}^{-1} (\boldsymbol{v}_{\texttt{item}} - \boldsymbol{\mu}_{\texttt{context}}) + \text{const.}
\end{aligned}
\tag{1}
$$

where $\texttt{context}$ denotes all the variables that the attention depends on. For example, "American engineers in the 18th century" or "American politicians in the 18th century" would be two contexts that include Benjamin Franklin but the two attentions would have very different shapes.

Compared to the (normalized) inner product used in previous work (Sukhbaatar et al., 2015; Graves et al., 2014) for content-based addressing, the Gaussian model has the additional control of the spread of the attention over items in the memory. As we show in Figure 1, we can view the conventional inner-product-based attention and the proposed Gaussian attention as addressing by an affine energy function and a quadratic energy function, respectively. By making the addressing mechanism more complex, we may represent many entities in a relatively low dimensional embedding space. Since knowledge bases are typically extremely sparse, it is more likely that we can afford to have a more complex attention model than a large embedding dimension.

We apply the proposed Gaussian attention model to question answering based on knowledge bases. At the high-level, the goal of the task is to learn the mapping from a question about objects in the knowledge base in natural language to a probability distribution over the entities. We use the scoring function (1) for both embedding the entities as vectors, and extracting the conditions mentioned in the question and taking a conjunction of them to score each candidate answer to the question.

The ability to compactly represent a set of objects makes the Gaussian attention model well suited for representing the uncertainty in a multiple-answer question (e.g., "who are the children of Abraham Lincoln?"). Moreover, traversal over the knowledge graph (see Guu et al., 2015) can be naturally handled by a series of Gaussian convolutions, which generalizes the addition of vectors. In fact, we model each relation as a Gaussian with mean and variance parameters. Thus a traversal on a relation corresponds to a translation in the mean and addition of the variances.

The proposed question answering model is able to handle not only the case where the answer to a question is associated with an atomic fact, which is called simple Q&A (Bordes et al., 2015), but also questions that require composition of relations (path queries in Guu et al. (2015)) and conjunction of queries. An example flow of how our model deals with a question "Who plays forward for Borussia Dortmund?" is shown in Figure 2 in Section 3.

This paper is structured as follows. In Section 2, we describe how the Gaussian scoring function (1) can be used to embed the entities in a knowledge base into a continuous vector space. We call our model TransGaussian because of its similarity to the TransE model proposed by Bordes et al. (2013). Then in Section 3, we describe our question answering model. In Section 4, we carry out experiments on WorldCup2014 dataset we collected. The dataset is relatively small but it allows us to evaluate not only simple questions but also path queries and conjunction of queries. The proposed TransGaussian embedding with the question answering model achieves significantly higher accuracy than the vanilla TransE embedding or TransE trained with compositional relations Guu et al. (2015) combined with the same question answering model.

## 2    KNOWLEDGE BASE EMBEDDING

In this section, we describe the proposed TransGaussian model based on the Gaussian attention model (1). While it is possible to train a network that computes the embedding in a single pass (Bordes et al., 2015) or over multiple passes (Li et al., 2015), it is more efficient to offload the embedding as a separate step for question answering based on a large static knowledge base.

### 2.1    THE TRANSGAUSSIAN MODEL

Let $\mathcal{E}$ be the set of entities and $\mathcal{R}$ be the set of relations. A knowledge base is a collection of triplets $(s, r, o)$, where we call $s \in \mathcal{E}$, $r \in \mathcal{R}$, and $o \in \mathcal{E}$, the subject, the relation, and the object of the triplet, respectively. Each triplet encodes a *fact*. For example, (Albert_Einstein, has_profession, theoretical_physicist). All the triplets given in a knowledge base are assumed to be true. However generally speaking a triplet may be true or false. Thus knowledge base embedding aims at training a model that predict if a triplet is true or not given some parameterization of the entities and relations (Bordes et al., 2011; 2013; Nickel et al., 2011; Socher et al., 2013; Wang et al., 2014).

In this paper, we associate a vector $\boldsymbol{v}_s \in \mathbb{R}^d$ with each entity $s \in \mathcal{E}$, and we associate each relation $r \in \mathcal{R}$ with two parameters, $\boldsymbol{\delta}_r \in \mathbb{R}^d$ and a positive definite symmetric matrix $\boldsymbol{\Sigma}_r \in \mathbb{R}^{d \times d}_{++}$.

Given subject $s$ and relation $r$, we can compute the score of an object $o$ to be in triplet $(s, r, o)$ using the Gaussian attention model as (1) with

$$\text{score}(s, r, o) = \log \phi(\boldsymbol{v}_o | \boldsymbol{\mu}_{\text{context}}, \boldsymbol{\Sigma}_{\text{context}}), \tag{2}$$

where $\boldsymbol{\mu}_{\text{context}} = \boldsymbol{v}_s + \boldsymbol{\delta}_r$, $\boldsymbol{\Sigma}_{\text{context}} = \boldsymbol{\Sigma}_r$. Note that if $\boldsymbol{\Sigma}_r$ is fixed to the identity matrix, we are modeling the relation of subject $\boldsymbol{v}_s$ and object $\boldsymbol{v}_o$ as a translation $\boldsymbol{\delta}_r$, which is equivalent to the TransE model (Bordes et al., 2013). We allow the covariance $\boldsymbol{\Sigma}_r$ to depend on the relation to handle one-to-many relations (e.g., profession_has_person relation) and capture the shape of the distribution of the set of objects that can be in the triplet. We call our model **TransGaussian** because of its similarity to TransE (Bordes et al., 2013).

**Parameterization**    For computational efficiency, we will restrict the covariance matrix $\boldsymbol{\Sigma}_r$ to be diagonal in this paper. Furthermore, in order to ensure that $\boldsymbol{\Sigma}_r$ is strictly positive definite, we employ the exponential linear unit (ELU, Clevert et al., 2015) and parameterize $\boldsymbol{\Sigma}_r$ as follows:

$$\boldsymbol{\Sigma}_r = \text{diag} \begin{pmatrix} \text{ELU}(m_{r,1})+1+\epsilon & & \\ & \ddots & \\ & & \text{ELU}(m_{r,d})+1+\epsilon \end{pmatrix}$$

where $m_{r,j}$ $(j = 1, \ldots, d)$ are the unconstrained parameters that are optimized during training and $\epsilon$ is a small positive value that ensure the positivity of the variance during numerical computation. The ELU is defined as

$$\text{ELU}(x) = \begin{cases} x, & x \geq 0, \\ \exp(x) - 1, & x < 0. \end{cases}$$

**Ranking loss**    Suppose we have a set of triplets $\mathcal{T} = \{(s_i, r_i, o_i)\}_{i=1}^N$ from the knowledge base. Let $\mathcal{N}(s, r)$ be the set of incorrect objects to be in the triplet $(s, r, \cdot)$.

Our objective function uses the ranking loss to measure the margin between the scores of true answers and those of false answers and it can be written as follows:

$$\min_{\substack{\{\boldsymbol{v}_e : e \in \mathcal{E}\}, \\ \{\boldsymbol{\delta}_r, \boldsymbol{M}_r, : r \in \bar{\mathcal{R}}\}}} \frac{1}{N} \sum_{(s,r,o) \in \mathcal{T}} \mathbb{E}_{t' \sim \mathcal{N}(s,r)} \left[ [\mu - \text{score}(s, r, o) + \text{score}(s, r, t')]_+ \right]$$

$$+ \lambda \left( \sum_{e \in \mathcal{E}} \|\boldsymbol{v}_e\|_2^2 + \sum_{r \in \bar{\mathcal{R}}} \left( \|\boldsymbol{\delta}_r\|_2^2 + \|\boldsymbol{M}_r\|_F^2 \right) \right), \tag{3}$$

where, $N = |\mathcal{T}|$, $\mu$ is the margin parameter and $\boldsymbol{M}_r$ denotes the diagonal matrix with $m_{r,j}$, $j = 1, \ldots, d$ on the diagonal; the function $[\cdot]_+$ is defined as $[x]_+ = \max(0, x)$. Here, we treat an inverse

relation as a separate relation and denote by $\bar{\mathcal{R}} = \mathcal{R} \cup \mathcal{R}^{-1}$ the set of all the relations including both relations in $\mathcal{R}$ and their inverse relations; a relation $\tilde{r}$ is the inverse relation of $r$ if $(s, \tilde{r}, o)$ implies $(o, r, s)$ and vice versa. Moreover, $\mathbb{E}_{t' \sim \mathcal{N}(s,r)}$ denotes the expectation with respect to the uniform distribution over the set of incorrect objects, which we approximate with 10 random samples in the experiments. Finally, the last terms are $\ell_2$ regularization terms for the embedding parameters.

## 2.2 COMPOSITIONAL RELATIONS

Guu et al. (2015) has recently shown that training TransE with *compositional relations* can make it competitive to more complex models, although TransE is much simpler compared to for example, neural tensor networks (NTN, Socher et al. (2013)) and TransH Wang et al. (2014). Here, a compositional relation is a relation that is composed as a series of relations in $\mathcal{R}$, for example, `grand_father_of` can be composed as first applying the `parent_of` relation and then the `father_of` relation, which can be seen as a traversal over a path on the knowledge graph.

TransGaussian model can naturally handle and propagate the uncertainty over such a chain of relations by convolving the Gaussian distributions along the path. That is, the score of an entity $o$ to be in the $\tau$-step relation $r_1/r_2/\cdots/r_\tau$ with subject $s$, which we denote by the triplet $(s, r_1/r_2/\cdots/r_\tau, o)$, is given as

$$\text{score}(s, r_1/r_2/\cdots/r_\tau, o) = \log \phi(\boldsymbol{v}_o | \boldsymbol{\mu}_{\text{context}}, \boldsymbol{\Sigma}_{\text{context}}), \tag{4}$$

with $\boldsymbol{\mu}_{\text{context}} = \boldsymbol{v}_s + \sum_{t=1}^{\tau} \boldsymbol{\delta}_{r_t}$, $\boldsymbol{\Sigma}_{\text{context}} = \sum_{t=1}^{\tau} \boldsymbol{\Sigma}_{r_t}$, where the covariance associated with each relation is parameterized in the same way as in the previous subsection.

**Training with compositional relations** Let $\mathcal{P} = \left\{ \left( s_i, r_{i_1}/r_{i_2}/\cdots/r_{i_{l_i}}, o_i \right) \right\}_{i=1}^{N'}$ be a set of randomly sampled paths from the knowledge graph. Here relation $r_{i_k}$ in a path can be a relation in $\mathcal{R}$ or an inverse relation in $\mathcal{R}^{-1}$. With the scoring function (4), the generalized training objective for compositional relations can be written identically to (3) except for replacing $\mathcal{T}$ with $\mathcal{T} \cup \mathcal{P}$ and replacing $N$ with $N' = |\mathcal{T} \cup \mathcal{P}|$.

## 3 QUESTION ANSWERING

Given a set of question-answer pairs, in which the question is phrased in natural language and the answer is an entity in the knowledge base, our goal is to train a model that learns the mapping from the question to the correct entity. Our question answering model consists of three steps, entity recognition, relation composition, and conjunction. We first identify a list of entities mentioned in the question (which is assumed to be provided by an oracle in this paper). If the question is "Who plays Forward for Borussia Dortmund?" then the list would be [`Forward`, `Borussia_Dortmund`]. The next step is to predict the path of relations on the knowledgegraph starting from each entity in the list extracted in the first step. In the above example, this will be (smooth versions of) `/Forward/position_played_by/` and `/Borussia_Dortmund/has_player/` predicted as series of Gaussian convolutions. In general, we can have multiple relations appearing in each path. Finally, we take a product of all the Gaussian attentions and renormalize it, which is equivalent to Bayes' rule with independent observations (paths) and a noninformative prior.

## 3.1 ENTITY RECOGNITION

We assume that there is an oracle that provides a list containing all the entities mentioned in the question, because (1) a domain specific entity recognizer can be developed efficiently (Williams et al., 2015) and (2) generally entity recognition is a challenging task and it is beyond the scope of this paper to show whether there is any benefit in training our question answering model jointly with a entity recognizer. We assume that the number of extracted entities can be different for each question.

## 3.2 RELATION COMPOSITION

We train a long short-term memory (LSTM, Hochreiter & Schmidhuber, 1997) network that emits an output $\boldsymbol{h}_t$ for each token in the input sequence. Then we compute the attention over the hidden

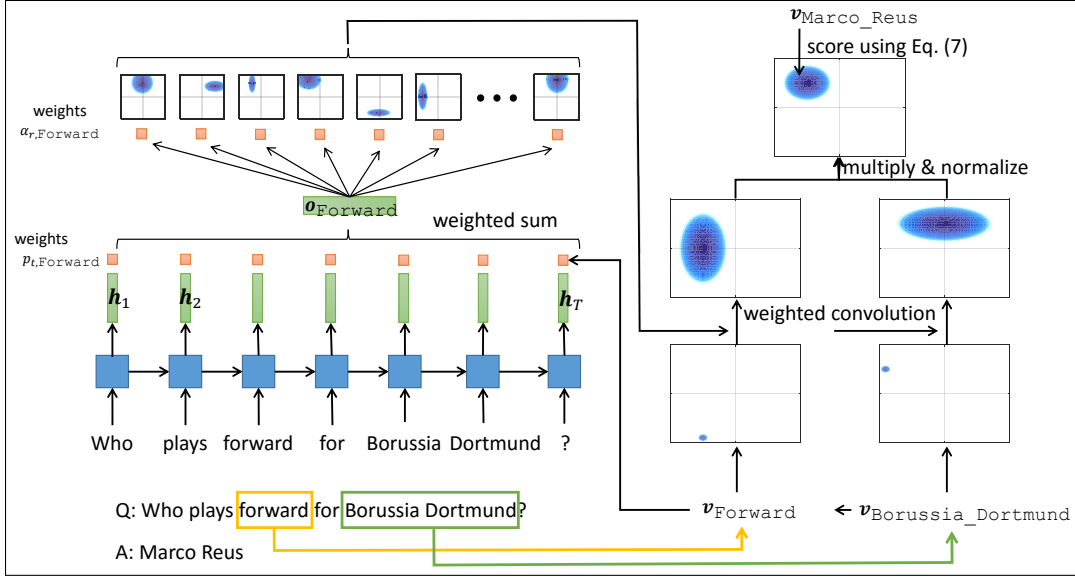

Figure 2: The input to the system is a question in natural language. Two entities `Forward` and `Borussia_Dortmund` are identified in the question and associated with point mass distributions centered at the corresponding entity vectors. An LSTM encodes the input into a sequence of output vectors of the same length. Then we take average of the output vectors weighted by attention $p_{t,e}$ for each recognized entity $e$ to predict the weight $\alpha_{r,e}$ for relation $r$ associated with entity $e$. We form a Gaussian attention over the entities for each entity $e$ by convolving the corresponding point mass with the (pre-trained) Gaussian embeddings of the relations weighted by $\alpha_{r,e}$ according to Eq. (6). The final prediction is produced by taking the product and normalizing the Gaussian attentions.

states for each recognized entity $e$ as

$$p_{t,e} = \text{softmax}\left(f(\boldsymbol{v}_e, \boldsymbol{h}_t)\right) \quad (t = 1, \ldots, T),$$

where $\boldsymbol{v}_e$ is the vector associated with the entity $e$. We use a two-layer perceptron for $f$ in our experiments, which can be written as follows:

$$f(\boldsymbol{v}_e, \boldsymbol{h}_t) = \boldsymbol{u}_f^\top \text{ReLU}\left(\boldsymbol{W}_{f,v}\boldsymbol{v}_e + \boldsymbol{W}_{f,h}\boldsymbol{h}_t + \boldsymbol{b}_1\right) + b_2,$$

where $\boldsymbol{W}_{f,v} \in \mathbb{R}^{L \times d}$, $\boldsymbol{W}_{f,h} \in \mathbb{R}^{L \times H}$, $\boldsymbol{b}_1 \in \mathbb{R}^L$, $\boldsymbol{u}_f \in \mathbb{R}^L$, $b_2 \in \mathbb{R}$ are parameters. Here $\text{ReLU}(x) = \max(0, x)$ is the rectified linear unit. Finally, softmax denotes softmax over the $T$ tokens.

Next, we use the weights $p_{t,e}$ to compute the weighted sum over the hidden states $\boldsymbol{h}_t$ as

$$\boldsymbol{o}_e = \sum\nolimits_{t=1}^{T} p_{t,e}\boldsymbol{h}_t. \tag{5}$$

Then we compute the weights $\alpha_{r,e}$ over all the relations as $\alpha_{r,e} = \text{ReLU}\left(\boldsymbol{w}_r^\top \boldsymbol{o}_e\right) \quad (\forall r \in \mathcal{R} \cup \mathcal{R}^{-1})$. Here the rectified linear unit is used to ensure the positivity of the weights. Note however that the weights should not be normalized, because we may want to use the same relation more than once in the same path. Making the weights positive also has the effect of making the attention sparse and interpretable because there is no cancellation.

For each extracted entity $e$, we view the extracted entity and the answer of the question to be the subject and the object in some triplet $(e, p, o)$, respectively, where the path $p$ is inferred from the question as the weights $\alpha_{r,e}$ as we described above. Accordingly, the score for each candidate answer $o$ can be expressed using (1) as:

$$\text{score}_e(\boldsymbol{v}_o) = \log \phi(\boldsymbol{v}_o | \boldsymbol{\mu}_{e,\alpha,\text{KB}}, \boldsymbol{\Sigma}_{e,\alpha,\text{KB}}) \tag{6}$$

with $\boldsymbol{\mu}_{e,\alpha,\text{KB}} = \boldsymbol{v}_e + \sum_{r \in \bar{\mathcal{R}}} \alpha_{r,e} \boldsymbol{\delta}_r$, $\boldsymbol{\Sigma}_{e,\alpha,\text{KB}} = \sum_{r \in \bar{\mathcal{R}}} \alpha_{r,e}^2 \boldsymbol{\Sigma}_r$, where $\boldsymbol{v}_e$ is the vector associated with entity $e$ and $\bar{\mathcal{R}} = \mathcal{R} \cup \mathcal{R}^{-1}$ denotes the set of relations including the inverse relations.

### 3.3 CONJUNCTION

Let $\mathcal{E}(q)$ be the set of entities recognized in the question $q$. The final step of our model is to take the conjunction of the Gaussian attentions derived in the previous step. This step is simply carried out by multiplying the Gaussian attentions as follows:

$$
\begin{aligned}
\text{score}(\boldsymbol{v}_o|\mathcal{E}(q), \Theta) &= \log \prod_{e \in \mathcal{E}(q)} \phi(\boldsymbol{v}_o|\boldsymbol{\mu}_{e,\alpha,\text{KB}}, \boldsymbol{\Sigma}_{e,\alpha,\text{KB}}) \\
&= -\frac{1}{2} \sum_{e \in \mathcal{E}(q)} \left(\boldsymbol{v}_o - \boldsymbol{\mu}_{e,\alpha,\text{KB}}\right)^\top \boldsymbol{\Sigma}_{e,\alpha,\text{KB}}^{-1} (\boldsymbol{v}_o - \boldsymbol{\mu}_{e,\alpha,\text{KB}}) + \text{const.,} \quad (7)
\end{aligned}
$$

which is again a (logarithm of) Gaussian scoring function, where $\boldsymbol{\mu}_{e,\alpha,\text{KB}}$ and $\boldsymbol{\Sigma}_{e,\alpha,\text{KB}}$ are the mean and the covariance of the Gaussian attention given in (6). Here $\Theta$ denotes all the parameters of the question-answering model.

### 3.4 TRAINING THE QUESTION ANSWERING MODEL

Suppose we have a knowledge base $(\mathcal{E}, \mathcal{R}, \mathcal{T})$ and a trained TransGaussian model $\left(\{v_e\}_{e \in \mathcal{E}}, \{(\boldsymbol{\delta}_r, \boldsymbol{\Sigma}_r)\}_{r \in \mathcal{R}}\right)$, where $\mathcal{R}$ is the set of all relations including the inverse relations. During training time, we assume the training set is a supervised question-answer pairs $\{(q_i, \mathcal{E}(q_i), a_i) : i = 1, 2, \ldots, m\}$. Here, $q_i$ is a question formulated in natural language, $\mathcal{E}(q_i) \subset \mathcal{E}$ is a set of knowledge base entities that appears in the question, and $a_i \in \mathcal{E}$ is the answer to the question. For example, on a knowledge base of soccer players, a valid training sample could be

("Who plays forward for Borussia Dortmund?", [Forward, Borussia_Dortmund], Marco_Reus).

Note that the answer to a question is not necessarily unique and we allow $a_i$ to be any of the true answers in the knowledge base. During test time, our model is shown $(q_i, \mathcal{E}(q_i))$ and the task is to find $a_i$. We denote the set of answers to $q_i$ by $A(q_i)$.

To train our question-answering model, we minimize the objective function

$$
\frac{1}{m} \sum_{i=1}^m \left( \mathop{\mathbb{E}}_{t' \sim \mathcal{N}(q_i)} \left[ [\mu - \text{score}(\boldsymbol{v}_{a_i}|\mathcal{E}(q_i), \Theta) + \text{score}(\boldsymbol{v}_{t'}|\mathcal{E}(q_i), \Theta)]_+ \right] + \nu \sum_{e \in \mathcal{E}(q_i)} \sum_{r \in \mathcal{R}} |\alpha_{r,e}| \right) + \lambda \|\Theta\|_2^2
$$

where $\mathbb{E}_{t' \sim \mathcal{N}(q_i)}$ is expectation with respect to a uniform distribution over of all incorrect answers to $q_i$, which we approximate with 10 random samples. We assume that the number of relations implied in a question is small compared to the total number of relations in the knowledge base. Hence the coefficients $\alpha_{r,e}$ computed for each question $q_i$ are regularized by their $\ell_1$ norms.

## 4 EXPERIMENTS

As a demonstration of the proposed framework, we perform question and answering on a dataset of soccer players. In this work, we consider two types of questions. A *path query* is a question that contains only one named entity from the knowledge base and its answer can be found from the knowledge graph by walking down a path consisting of a few relations. A *conjunctive query* is a question that contains more than one entities and the answer is given as the conjunction of all path queries starting from each entity. Furthermore, we experimented on a knowledge base completion task with TransGaussian embeddings to test its capability of generalization to unseen fact. Since knowledge base completion is not the main focus of this work, we include the results in the Appendix.

### 4.1 WORLDCUP2014 DATASET

We build a knowledge base of football players that participated in FIFA World Cup 2014 [1]. The original dataset consists of players' information such as nationality, positions on the field and ages etc. We picked a few attributes and constructed 1127 entities and 6 atomic relations. The entities include 736 players, 297 professional soccer clubs, 51 countries, 39 numbers and 4 positions. And the six atomic relations are

---

[1]The original dataset can be found at https://datahub.io/dataset/fifa-world-cup-2014-all-players.

`plays_in_club`: PLAYER → CLUB,           `plays_position`: PLAYER → POSITION,
`is_aged`: PLAYER → NUMBER,               `wears_number`[2]: PLAYER → NUMBER,
`plays_for_country`: PLAYER → COUNTRY,    `is_in_country`: CLUB → COUNTRY,

where PLAYER, CLUB, NUMBER, etc, denote the type of entities that can appear as the left or right argument for each relation. Some relations share the same type as the right argument, e.g., `plays_for_country` and `is_in_country`.

Given the entities and relations, we transformed the dataset into a set of 3977 triplets. A list of sample triplets can be found in the Appendix. Based on these triplets, we created two sets of question answering tasks which we call *path query* and *conjunctive query* respectively. The answer of every question is always an entity in the knowledge base and a question can involve one or two triplets. The questions are generated as follows.

**Path queries.**   Among the paths on the knowledge graph, there are some natural composition of relations, e.g., `plays_in_country` (PLAYER → COUNTRY) can be decomposed as the composition of `plays_in_club` (PLAYER→ CLUB) and `is_in_country` (CLUB → COUNTRY). In addition to the atomic relations, we manually picked a few meaningful compositions of relations and formed *query templates*, which takes the form "find $e \in \mathcal{E}$, such that $(s, p, e)$ is true", where $s$ is the subject and $p$ can be an atomic relation or a path of relations. To formulate a set of path-based question-answer pairs, we manually created one or more *question templates* for every query template (see Table 5) Then, for a particular instantiation of a query template with subject and object entities, we randomly select a question template to generate a question given the subject; the object entity becomes the answer of the question. See Table 6 for the list of composed relations, sample questions, and answers. Note that all atomic relations in this dataset are many-to-one while these composed relations can be one-to-many or many-to-many as well.

**Conjunctive queries.**   To generate question-and-answer pairs of conjunctive queries, we first picked three pairs of relations and used them to create query templates of the form "Find $e \in \mathcal{E}$, such that both $(s_1, r_1, e)$ and $(s_2, r_2, e)$ are true." (see Table 5). For a pair of relations $r_1$ and $r_2$, we enumerated all pairs of entities $s_1$, $s_2$ that can be their subjects and formulated the corresponding query in natural language using question templates as in the same way as path queries. See Table 7 for a list of sample questions and answers.

As a result, we created 8003 question-and-answer pairs of path queries and 2208 pairs of conjunctive queries which are partitioned into train / validation / test subsets. We refer to Table 1 for more statistics about the dataset. Templates for generating the questions are list in Table 5.

## 4.2   Experimental setup

To perform question and answering under our proposed framework, we first train the TransGaussian model on WorldCup2014 dataset. In addition to the atomic triplets, we randomly sampled 50000 paths with length 1 or 2 from the knowledge graph and trained a TransGaussian model compositionally as described in Set 2.2. An inverse relation is treated as a separate relation. Following the naming convention from Guu et al. (2015), we denote this trained embedding by *TransGaussian (COMP)*. We found that the learned embedding possess some interesting properties. Some dimensions of the embedding space dedicate to represent a particular relation. Players are clustered by their attributes when entities' embeddings are projected to the corresponding lower dimensional subspaces. We elaborate and illustrate such properties in the Appendix.

**Baseline methods**   We also trained a TransGaussian model only on the atomic triplets and denote such a model by *TransGaussian (SINGLE)*. Since no inverse relation was involved when *TransGaussian (SINGLE)* was trained, to use this embedding in question answering tasks, we represent the inverse relations as follows: for each relation $r$ with mean $\boldsymbol{\delta}_r$ and variance $\boldsymbol{\Sigma}_r$, we model its inverse $r^{-1}$ as a Gaussian attention with mean $-\boldsymbol{\delta}_r$ and variance equal to $\boldsymbol{\Sigma}_r$.

We also trained TransE models on WorldCup2014 dataset by using the code released by the authors of Guu et al. (2015). Likewise, we use *TransE (SINGLE)* to denote the model trained with atomic triplets only and use *TransE (COMP)* to denote the model trained with the union of triplets and paths. Note that TransE can be considered as a special case of TransGaussian where the variance matrix is the identity and hence, the scoring formula Eq. (7) is applicable to TransE as well.

**Training configurations** For all models, dimension of entity embeddings was set to 30. The hidden size of LSTM was set to 80. Word embeddings were trained jointly with the question answering model and dimension of word embedding was set to 40. We employed Adam (Kingma & Ba, 2014) as the optimizer. All parameters were tuned on the validation set. Under the same setting, we experimented with two cases: first, we trained models for path queries and conjunctive queries separately; Furthermore, we trained a single model that addresses both types queries. We present the results of the latter case in the next subsection while the results of the former are included in the Appendix.

**Evaluation metrics** During test time, our model receives a question in natural language and a list of knowledge base entities contained in the question. Then it predicts the mean and variance of a Gaussian attention formulated in Eq. (7) which is expected to capture the distribution of all positive answers. We rank all entities in the knowledge base by their scores under this Gaussian attention. Next, for each entity which is a correct answer, we check its rank relative to all incorrect answers and call this rank the filtered rank. For example, if a correct entity is ranked above all negative answers except for one, it has filtered rank two. We compute this rank for all true answers and report *mean filtered rank* and *H@1* which is the percentage of true answers that have filtered rank 1.

## 4.3 EXPERIMENTAL RESULTS

We present the results of joint learning in Table 2. These results show that TransGaussian works better than TransE in general. In fact, *TransGaussian (COMP)* achieved the best performance in almost all aspects. Most notably, it achieved the highest H@1 rates on challenging questions such as "where is the club that edin dzeko plays for?" (#11, composition of two relations) and "who are the defenders on german national team?" (#14, conjunction of two queries).

The same table shows that TransGaussian benefits remarkably from compositional training. For example, compositional training improved TransGaussian's H@1 rate by near 60% in queries on players from a given countries (#8) and queries on players who play a particular position (#9). It also boosted TransGaussian's performance on all conjunctive quries (#13–#15) significantly.

To understand *TransGaussian (COMP)*'s weak performance on answering queries on the professional football club located in a given country (#10) and queries on professional football club that has players from a particular country (#12), we tested its capability of modeling the composed relation by feeding the correct relations and subjects during test time. It turns out that these two relations were not modeled well by *TransGaussian (COMP)* embedding, which limits its performance in question answering. (See Table 8 in the Appendix for quantitative evaluations.) The same limit was found in the other three embeddings as well.

Note that all the models compared in Table 2 uses the proposed Gaussian attention model because TransE is the special case of TransGaussian where the variance is fixed to one. Thus the main differences are whether the variance is learned and whether the embedding was trained compositionally. Finally, we refer to Table 9 and 10 in the Appendix for experimental results of models trained on path and conjunctive queries separately.

Table 1: Some statistics of the WorldCup2014 dataset.

| # entity | # atomic relations | # atomic triplets | # path query Q&A ( train / validation / test ) | # conjunctive query Q&A ( train / validation / test ) |
|---|---|---|---|---|
| 1127 | 6 | 3977 | 5620 / 804 / 1579 | 1564 / 224 / 420 |

## 5 RELATED WORK

The work of Vilnis & McCallum (2014) is similar to our Gaussian attention model. They discuss many advantages of the Gaussian embedding; for example, it is arguably a better way of handling asymmetric relations and entailment. However the work was presented in the word2vec (Mikolov et al., 2013)-style word embedding setting and the Gaussian embedding was used to capture the diversity in the meaning of a word. Our Gaussian attention model extends their work to a more general setting in which any memory item can be addressed through a concept represented as a Gaussian distribution over the memory items.

Table 2: Results of joint learning with path queries and conjunction queries on WorldCup2014.

| # | Sample question | TransE (SINGLE) | | TransE (COMP) | | TransGaussian (SINGLE) | | TransGaussian (COMP) | |
|---|---|---|---|---|---|---|---|---|---|
| | | H@1(%) | Mean Filtered Rank | H@1(%) | Mean Filtered Rank | H@1(%) | Mean Filtered Rank | H@1(%) | Mean Filtered Rank |
| 1 | which club does alan pulido play for? | 88.59 | 1.18 | 91.95 | 1.11 | 96.64 | 1.04 | **98.66** | **1.01** |
| 2 | what position does gonzalo higuain play? | *100.00* | *1.00* | 98.11 | 1.03 | 98.74 | 1.01 | *100.00* | *1.00* |
| 3 | how old is samuel etoo? | 67.11 | 1.44 | 90.79 | 1.13 | 94.74 | 1.08 | **97.37** | **1.04** |
| 4 | what is the jersey number of mario balotelli? | 45.00 | 1.89 | 83.57 | 1.22 | 97.14 | 1.03 | **99.29** | **1.01** |
| 5 | which country is thomas mueller from ? | 94.40 | 1.06 | 94.40 | 1.06 | 96.80 | 1.04 | **98.40** | **1.02** |
| 6 | which country is the soccer team fc porto based in ? | *98.48* | *1.02* | *98.48* | *1.02* | 93.94 | 1.06 | 95.45 | 1.05 |
| 7 | who plays professionally at liverpool fc? | 95.12 | 1.10 | 90.24 | 1.20 | **98.37** | *1.04* | 96.75 | *1.04* |
| 8 | which player is from iran? | 89.86 | 1.51 | 76.81 | 2.07 | 38.65 | 2.96 | **99.52** | **1.00** |
| 9 | name a player who plays goalkeeper? | 98.96 | 1.01 | 69.79 | 1.82 | 42.71 | 5.52 | **100.00** | **1.00** |
| 10 | which soccer club is based in mexico? | 22.03 | 13.94 | **30.51** | **8.84** | 6.78 | 10.66 | 16.95 | 21.14 |
| 11 | where is the club that edin dzeko plays for ? | 52.63 | 3.88 | 57.24 | 2.10 | 47.37 | 2.27 | **78.29** | **1.41** |
| 12 | name a soccer club that has a player from australia ? | 30.43 | 12.08 | **33.70** | **11.47** | 13.04 | 11.64 | 19.57 | 17.57 |
| | Overall (Path Query) | 74.16 | 3.11 | 77.39 | **2.56** | 69.54 | 3.02 | **85.94** | 3.52 |
| 13 | who plays forward for fc barcelona? | 97.55 | 1.06 | 76.07 | 1.66 | 93.25 | 1.24 | **98.77** | **1.02** |
| 14 | who are the defenders on german national team? | 95.93 | 1.06 | 69.92 | 2.33 | 65.04 | 2.04 | **100.00** | **1.00** |
| 15 | which player in ssc napoli is from argentina? | 88.81 | 1.17 | 76.12 | 1.76 | 88.81 | 1.35 | **97.76** | **1.03** |
| | Overall (Conj. Query) | 94.29 | 1.09 | 74.29 | 1.89 | 83.57 | 1.51 | **98.81** | **1.02** |

Bordes et al. (2014; 2015) proposed a question-answering model that embeds both questions and their answers to a common continuous vector space. Their method in Bordes et al. (2015) can combine multiple knowledge bases and even generalize to a knowledge base that was not used during training. However their method is limited to the simple question answering setting in which the answer of each question associated with a triplet in the knowledge base. In contrast, our method can handle both composition of relations and conjunction of conditions, which are both naturally enabled by the proposed Gaussian attention model.

Neelakantan et al. (2015a) proposed a method that combines relations to deal with compositional relations for knowledge base completion. Their key technical contribution is to use recurrent neural networks (RNNs) to *encode* a chain of relations. When we restrict ourselves to path queries, question answering can be seen as a sequence transduction task (Graves, 2012; Sutskever et al., 2014) in which the input is text and the output is a series of relations. If we use RNNs as a *decoder*, our model would be able to handle non-commutative composition of relations, which the current weighted convolution cannot handle well. Another interesting connection to our work is that they take the maximum of the inner-product scores (see also Weston et al., 2013; Neelakantan et al., 2015b), which are computed along multiple paths connecting a pair of entities. Representing a set as a collection of vectors and taking the maximum over the inner-product scores is a natural way to represent a set of memory items. The Gaussian attention model we propose in this paper, however, has the advantage of differentiability and composability.

## 6 CONCLUSION

In this paper, we have proposed the Gaussian attention model which can be used in a variety of contexts where we can assume that the distance between the memory items in the latent space is compatible with some notion of semantics. We have shown that the proposed Gaussian scoring function can be used for knowledge base embedding achieving competitive accuracy. We have also shown that our embedding model can naturally propagate uncertainty when we compose relations together. Our embedding model also benefits from compositional training proposed by Guu et al. (2015). Furthermore, we have demonstrated the power of the Gaussian attention model in a challenging question answering problem which involves both composition of relations and conjunction of queries. Future work includes experiments on natural question answering datasets and end-to-end training including the entity extractor.

ACKNOWLEDGMENTS

The authors would like to thank Daniel Tarlow, Nate Kushman, and Kevin Gimpel for valuable discussions.

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

## A  WORDCUP2014 DATASET

Table 3: Sample atomic triplets.

| Subject | Relation | Object |
|---------|----------|--------|
| david_villa | plays_for_country | spain |
| lionel_messi | plays_in_club | fc_barcelona |
| antoine_griezmann | plays_position | forward |
| cristiano_ronaldo | wears_number | 7 |
| fulham_fc | is_in_country | england |
| lukas_podolski | is_aged | 29 |

Table 4: Statistics of the WorldCup2014 dataset.

| | |
|---|---|
| # entity | 1127 |
| # atomic relations | 6 |
| # atomic triplets | 3977 |
| # relations (atomic and compositional) in path queries | 12 |
| # question and answer pairs in path queries ( train / validation / test ) | 5620 / 804 / 1579 |
| # types of questions in conjunctive queries | 3 |
| # question and answer pairs in conjunctive queries ( train / validation / test ) | 1564 / 224 / 420 |
| size of vocabulary | 1781 |

Table 5: Templates of questions. In the table, (player), (club), (position) are placeholders of named entities with associated type. (country_1) is a placeholder for a country name while (country_2) is a placeholder for the adjectival form of a country.

| # | Query template | Question template |
|---|----------------|-------------------|
| 1 | Find $e \in \mathcal{E}$: ( (player), plays_in_club, $e$) is true | which club does (player) play for ? <br> which professional football team does (player) play for ? <br> which football club does (player) play for ? |
| 2 | Find $e \in \mathcal{E}$: ((player), plays_position, $e$) is true | what position does (player) play ? |
| 3 | Find $e \in \mathcal{E}$: ((player), is_aged, $e$) is true | how old is (player) ? <br> what is the age of (player) ? |
| 4 | Find $e \in \mathcal{E}$: ((player), wears_number, $e$) is true | what is the jersey number of (player) ? <br> what number does (player) wear ? |
| 5 | Find $e \in \mathcal{E}$: ((player), plays_for_country, $e$) is true | what is the nationality of (player) ? <br> which national team does (player) play for ? <br> which country is (player) from ? |
| 6 | Find $e \in \mathcal{E}$: ((club), is_in_country, $e$) is true | which country is the soccer team (club) based in ? |
| 7 | Find $e \in \mathcal{E}$: ((club), plays_in_club$^{-1}$, $e$) is true | name a player from (club) ? <br> who plays at the soccer club (club) ? <br> who is from the professional football team (club) ? <br> who plays professionally at (club) ? |
| 8 | Find $e \in \mathcal{E}$: ((country_1), plays_for_country$^{-1}$, $e$) is true | which player is from (country_1) ? <br> name a player from (country_1) ? <br> who is from (country_1) ? <br> who plays for the (country_1) national football team ? |
| 9 | Find $e \in \mathcal{E}$: ((position), plays_position$^{-1}$, $e$) is true | name a player who plays (position) ? <br> who plays (position) ? |
| 10 | Find $e \in \mathcal{E}$: ((country_1), is_in_country$^{-1}$, $e$) is true | which soccer club is based in (country_1) ? <br> name a soccer club in (country_1) ? |
| 11 | Find $e \in \mathcal{E}$: ((player), plays_in_club / is_in_country, $e$) is true | which country does (player) play professionally in ? <br> where is the football club that (player) plays for ? |
| 12 | Find $e \in \mathcal{E}$: ((country_1), plays_for_country$^{-1}$ / plays_in_club, $e$) is true | which professional football team do players from (country_1) play for ? <br> name a soccer club that has a player from (country_1) ? <br> which professional football team has a player from (country_1) ? |
| 13 | Find $e \in \mathcal{E}$: ((position), plays_position$^{-1}$, $e$) is true and ((club), plays_in_club$^{-1}$, $e$) is true | who plays (position) for (club)? <br> who are the (position) at (club) ? <br> name a (position) that plays for (club) ? |
| 14 | Find $e \in \mathcal{E}$: ((position), plays_position$^{-1}$, $e$) is true and ((country_1), plays_for_country$^{-1}$, $e$) is true | who plays (position) for (country_1) ? <br> who are the (position) on (country_1) national team ? <br> name a (position) from (country_1) ? <br> which (country_2) footballer plays (position) ? <br> name a (country_2) (position) ? |
| 15 | Find $e \in \mathcal{E}$: ((club), plays_in_club$^{-1}$, $e$) is true and ((country_1), plays_for_country$^{-1}$, $e$) is true | who are the (country_2) players at (club) ? <br> which (country_2) footballer plays for (club) ? <br> name a (country_2) player at (club) ? <br> which player in (club) is from (country_1) ? |

Table 6: (Composed) relations and sample questions in path queries.

| # | Relation | Type | Sample question | Sample answer |
|---|---|---|---|---|
| 1 | plays_in_club | many-to-one | which club does alan pulido play for ?<br>which professional football team does klaas jan huntelaar play for ? | tigres_uanl<br>fc_schalke_04 |
| 2 | plays_position | many-to-one | what position does gonzalo higuain play ? | ssc_napoli |
| 3 | is_aged | many-to-one | how old is samuel etoo ?<br>what is the age of luis suarez ? | 33<br>27 |
| 4 | wears_number | many-to-one | what is the jersey number of mario balotelli ?<br>what number does shinji okazaki wear ? | 9<br>9 |
| 5 | plays_for_country | many-to-one | which country is thomas mueller from ?<br>what is the nationality of helder postiga ? | germany<br>portugal |
| 6 | is_in_country | many-to-one | which country is the soccer team fc porto based in ? | portugal |
| 7 | plays_in_club$^{-1}$ | one-to-many | who plays professionally at liverpool fc ?<br>name a player from as roma ? | steven_gerrard<br>miralem_pjanic |
| 8 | plays_for_country$^{-1}$ | one-to-many | which player is from iran ?<br>name a player from italy ? | masoud_shojaei<br>daniele_de_rossi |
| 9 | plays_position$^{-1}$ | one-to-many | name a player who plays goalkeeper ?<br>who plays forward ? | gianluiqi_buffon<br>raul_jimenez |
| 10 | is_in_country$^{-1}$ | one-to-many | which soccer club is based in mexico ?<br>name a soccer club in australia ? | cruz_azul_fc<br>melbourne_victory_fc |
| 11 | plays_in_club / is_in_country | many-to-one | where is the club that edin dzeko plays for ?<br>which country does sime vrsaljko play professionally in ? | england<br>italy |
| 12 | plays_for_country$^{-1}$/plays_in_club | many-to-many | name a soccer club that has a player from australia ?<br>name a soccer club that has a player from spain ? | crystal_palace_fc<br>fc_barcelona |

Table 7: Conjunctive queries and sample questions.

| # | Relations | Sample questions | Entities in questions | Sample answer |
|---|---|---|---|---|
| 13 | plays_position$^{-1}$<br>and<br>plays_in_club$^{-1}$ | who plays forward for fc barcelona ?<br>who are the midfielders at fc bayern muenchen ? | forward , fc_barcelona<br>midfielder, fc_bayern_muenchen | lionel_messi<br>toni_kroos |
| 14 | plays_position$^{-1}$<br>and<br>plays_for_country$^{-1}$ | who are the defenders on german national team ?<br>which mexican footballer plays forward ? | defender , germany<br>defender , mexico | per_mertesacker<br>raul_jimenez |
| 15 | plays_in_club$^{-1}$<br>and<br>plays_for_country$^{-1}$ | which player in paris saint-germain fc is from argentina ?<br>who are the korean players at beijing guoan ? | paris_saint-germain_fc , argentina<br>beijing_guoan , korea | ezequiel_lavezzi<br>ha_daesung |

# B TRANSGAUSSIAN EMBEDDING OF WORLDCUP2014

We trained our TransGaussian model on triplets and paths from WorldCup2014 dataset and illustrated the embeddings in Fig 3 and 4. Recall that we modeled every relation as a Gaussian with diagonal covariance matrix. Fig 3 shows the learned variance parameters of different relations. Each row corresponds to the variances of one relation. Columns are permuted to reveal the block structure. From this figure, we can see that every relation has a small variance in two or more dimensions. This implies that the coordinates of the embedding space are partitioned into semantically coherent clusters each of which represent a particular attribute of a player (or a football club). To verify this further, we picked the two coordinates in which a relation (e.g. plays_position) has the least variance and projected the embedding of all valid subjects and objects (e.g. players and positions) of the relation to this 2 dimensional subspace. See Fig. 4. The relation between the subjects and the objects are simply translation in the projection when the corresponding subspace is two dimensional (e.g., plays_position relation in Fig. 4 (a)). The same is true for other relations that requires larger dimension but it is more challenging to visualize in two dimensions. For relations that have a large number of unique objects, we only plotted for the eight objects with the most subjects for clarity of illustration.

Furthermore, in order to elucidate whether we are limited by the capacity of the TransGaussian embedding or the ability to decode question expressed in natural language, we evaluated the test question-answer pairs using the TransGaussian embedding composed according to the ground-truth relations and entities. The results were evaluated with the same metrics as in Sec. 4.3. This estimation is conducted for TransE embeddings as well. See Table 8 for the results. Compared to Table 2, the accuracy of TransGaussian (COMP) is higher on the atomic relations and path queries but lower on conjunctive queries. This is natural because when the query is simple there is not much room for the question-answering network to improve upon just combining the relations according to the ground truth relations, whereas when the query is complex the network could combine the embedding in a more creative way to overcome its limitation. In fact, the two queries (#10 and #12) that TransGaussian (COMP) did not perform well in Table 2 pertain to a single relation is_in_country$^{-1}$ (#10) and a composition of two relations plays_for_country$^{-1}$ / plays_in_club (#12). The performance of the two queries were low even when the ground truth

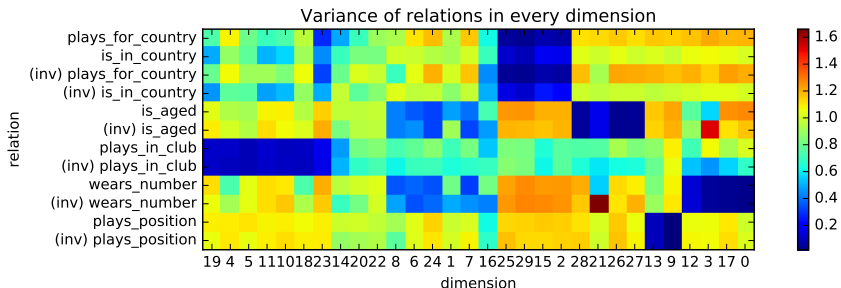

Figure 3: Variance of each relation. Each row shows the diagonal values in the variance matrix associated with a relation. Columns are permuted to reveal the block structure.

Table 8: Evaluation of embeddings. We evaluate the embeddings by feeding the correct entities and relations from a path or conjunctive query to an embedding model and using its scoring function to retrieve the answers from the embedded knowledge base.

| # | Relation | TransE (SINGLE) | | TransE (COMP) | | TransGaussian (SINGLE) | | TransGaussian (COMP) | |
|---|---|---|---|---|---|---|---|---|---|
| | | H@1(%) | Mean Filtered Rank | H@1(%) | Mean Filtered Rank | H@1(%) | Mean Filtered Rank | H@1(%) | Mean Filtered Rank |
| 1 | plays_in_club | 75.54 | 1.38 | 93.48 | 1.09 | **99.86** | **1.00** | 98.51 | 1.02 |
| 2 | plays_position | 96.33 | 1.04 | 94.02 | 1.09 | 98.37 | 1.02 | **100.00** | **1.00** |
| 3 | is_aged | 55.03 | 1.69 | 91.44 | 1.12 | 96.88 | 1.03 | **100.00** | **1.00** |
| 4 | wears_number | 38.86 | 2.09 | 78.67 | 1.32 | 95.92 | 1.04 | **100.00** | **1.00** |
| 5 | plays_for_country | 71.60 | 1.39 | 94.84 | 1.10 | 99.32 | 1.01 | **100.00** | **1.00** |
| 6 | is_in_country | 98.32 | 1.03 | 99.66 | 1.00 | 99.33 | 1.01 | **100.00** | **1.00** |
| 7 | plays_in_club$^{-1}$ | 87.50 | 1.46 | 83.42 | 1.45 | 94.70 | 1.07 | **97.42** | **1.03** |
| 8 | plays_for_country$^{-1}$ | 82.47 | 1.68 | 68.21 | 3.37 | 25.27 | 5.66 | **98.78** | **1.02** |
| 9 | plays_position$^{-1}$ | **100.00** | **1.00** | 75.54 | 1.60 | 13.59 | 24.35 | 98.78 | 1.02 |
| 10 | is_in_country$^{-1}$ | 23.11 | 26.92 | **23.48** | **23.27** | 8.32 | 130.59 | 19.41 | 83.61 |
| 11 | plays_in_club / is_in_country | 20.24 | 7.05 | 58.29 | 1.98 | 46.88 | 2.99 | **80.16** | **1.38** |
| 12 | plays_for_country$^{-1}$/plays_in_club | 25.32 | 22.27 | **27.73** | **10.04** | 19.04 | 35.59 | 20.15 | 33.01 |
| | Overall (Path relations) | 64.64 | 5.09 | 75.02 | **3.59** | 67.22 | 14.87 | **86.73** | 8.79 |
| 13 | plays_position$^{-1}$ and plays_in_club$^{-1}$ | 91.85 | 1.20 | 69.97 | 1.82 | 77.45 | 1.83 | **95.38** | **1.06** |
| 14 | plays_position$^{-1}$ and plays_for_country$^{-1}$ | 91.71 | 1.23 | 66.71 | 2.85 | 51.49 | 4.88 | **97.83** | **1.05** |
| 15 | plays_in_club$^{-1}$ and is_in_country$^{-1}$ | 88.59 | 1.20 | 73.37 | 1.80 | 83.42 | 1.34 | **94.70** | **1.08** |
| | Overall (Conj. relations) | 90.72 | 1.21 | 70.02 | 2.16 | 70.79 | 2.68 | **95.97** | **1.06** |

relations were given, which indicates that the TransGaussian embedding rather than the question-answering network is the limiting factor.

## C  KNOWLEDGE BASE COMPLETION

Knowledge base completion has been a common task for testing knowledge base models on their ability of generalizing to unseen facts. Here, we apply our TransGaussian model to a knowledge completion task and show that it has competitive performance.

We tested on the subset of WordNet released by Guu et al. (2015). The atomic triplets in this dataset was originally created by Socher et al. (2013) and Guu et al. (2015) added path queries that were randomly sampled from the knowledge graph. We build our TransGaussian model by training on these triplets and paths and tested our model on the same link prediction task as done by Socher et al. (2013); Guu et al. (2015).

As done by Guu et al. (2015), we trained *TransGaussian (SINGLE)* with atomic triplets only and trained *TransGaussian (COMP)* with the union of atomic triplets and paths. We did not incorporate

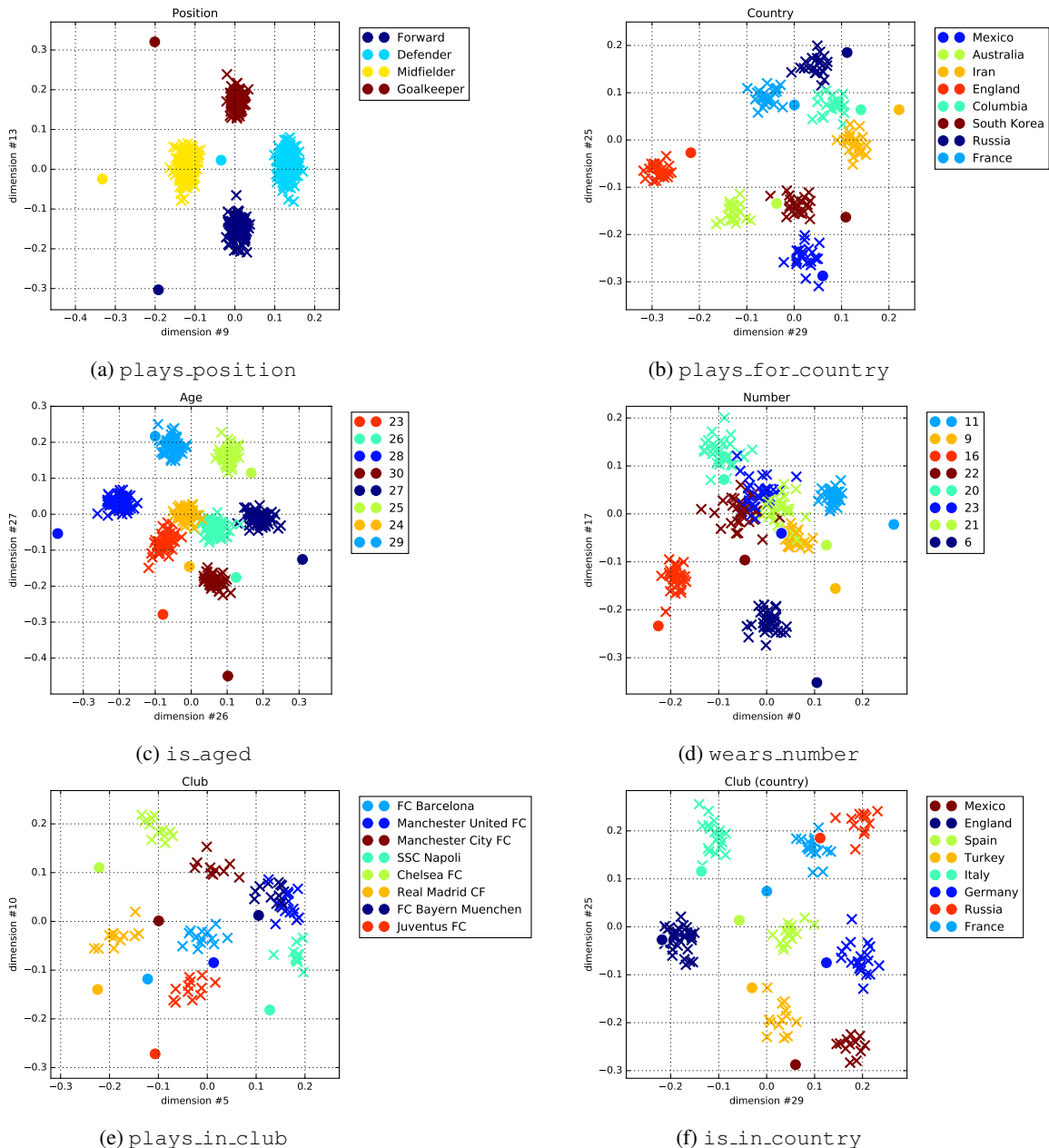

Figure 4: TransGaussian entity embeddings. Crosses are the subjects and circles are the objects of a relation. Specifically, crosses are players in (a)-(e) and professional football clubs in (f).

word embedding in this task and each entity is assigned its individual vector. Without getting parameters tuned too much, *TransGaussian (COMP)* obtained accuracy comparable to *TransE (COMP)*. See Table 11.

Table 9: Experimental results of path queries on WorldCup2014.

| # | Relation and sample question | TransE (SINGLE) | | TransE (COMP) | | TransGaussian (SINGLE) | | TransGaussian (COMP) | |
|---|---|---|---|---|---|---|---|---|---|
| | | H@1(%) | Mean Filtered Rank | H@1(%) | Mean Filtered Rank | H@1(%) | Mean Filtered Rank | H@1(%) | Mean Filtered Rank |
| 1 | plays_in_club (which club does alan pulido play for?) | 90.60 | 1.12 | 92.62 | 1.11 | 96.64 | *1.03* | **97.99** | *1.03* |
| 2 | plays_position (what position does gonzalo higuain play?) | *100.00* | *1.00* | 98.11 | 1.02 | 98.74 | 1.01 | *100.00* | *1.00* |
| 3 | is_aged (how old is samuel etoo?) | 81.58 | 1.30 | 92.11 | 1.10 | 96.05 | 1.04 | **100.00** | **1.00** |
| 4 | wears_number (what is the jersey number of mario balotelli?) | 44.29 | 1.88 | 85.71 | 1.19 | 96.43 | 1.04 | **100.00** | **1.00** |
| 5 | plays_for_country (which country is thomas mueller from ?) | 97.60 | 1.02 | 94.40 | 1.11 | 98.40 | 1.02 | **99.20** | **1.01** |
| 6 | is_in_country (which country is the soccer team fc porto based in ?) | *98.48* | *1.02* | *98.48* | *1.02* | 93.94 | 1.08 | *98.48* | *1.02* |
| 7 | plays_in_club$^{-1}$ (who plays professionally at liverpool fc?) | 95.12 | 1.08 | 86.99 | 1.38 | *96.75* | *1.03* | *96.75* | *1.03* |
| 8 | plays_for_country$^{-1}$ (which player is from iran?) | 81.16 | 1.61 | 72.46 | 2.36 | 40.58 | 3.19 | **93.24** | **1.48** |
| 9 | plays_position$^{-1}$ (name a player who plays goalkeeper?) | **100.00** | **1.00** | 30.21 | 2.30 | 55.21 | 5.09 | 85.42 | 1.15 |
| 10 | is_in_country$^{-1}$ (which soccer club is based in mexico?) | **24.58** | 11.47 | 23.73 | 10.07 | 5.08 | **9.18** | 17.80 | 20.10 |
| 11 | plays_in_club / is_in_country (where is the club that edin dzeko plays for ?) | 48.68 | 4.24 | 62.50 | 2.07 | 48.03 | 2.41 | **76.97** | **1.50** |
| 12 | plays_for_country$^{-1}$ / plays_in_club (name a soccer club that has a player from australia ?) | **34.78** | **9.49** | 30.43 | 11.26 | 6.52 | 9.88 | 16.30 | 20.27 |
| | Overall | 74.92 | 2.80 | 74.35 | **2.71** | 70.17 | 2.82 | **84.42** | 3.68 |

Table 10: Experimental results of conjunctive queries on WorldCup2014.

| # | Relation and sample question | TransE (SINGLE) | | TransE (COMP) | | TransGaussian (SINGLE) | | TransGaussian (COMP) | |
|---|---|---|---|---|---|---|---|---|---|
| | | H@1(%) | Mean Filtered Rank | H@1(%) | Mean Filtered Rank | H@1(%) | Mean Filtered Rank | H@1(%) | Mean Filtered Rank |
| 13 | plays_position$^{-1}$ and plays_in_club$^{-1}$ (who plays forward for fc barcelona?) | 94.48 | 1.10 | 71.17 | 1.77 | 87.12 | 1.37 | **98.77** | **1.02** |
| 14 | plays_position$^{-1}$ and plays_for_country$^{-1}$ (who are the defenders on german national team?) | 95.93 | 1.08 | 76.42 | 2.50 | 64.23 | 2.02 | **100.00** | **1.00** |
| 15 | plays_in_club$^{-1}$ and is_in_country$^{-1}$ (which player in ssc napoli is from argentina?) | 91.79 | 1.13 | 75.37 | 1.75 | 88.06 | 1.37 | **94.03** | **1.07** |
| | Overall | 94.05 | 1.11 | 74.05 | 1.97 | 80.71 | 1.56 | **97.62** | **1.03** |

Table 11: Accuracy of knowledge base completion on WordNet.

| Model | Accuracy (%) |
|---|---|
| TransE (SINGLE) | 68.5 |
| TransE (COMP) | 80.3 |
| TransGaussian (SINGLE) | 58.4 |
| TransGaussian (COMP) | 76.4 |

