# Peer review of "Gaussian Attention Model and Its Application to Knowledge Base Embedding and Question Answering"

_ICLR 2017 — rejected_

[Official Review · AnonReviewer3 · rating 4 · confidence 3 · 16 Dec 2016]
**No Title**

SUMMARY.

The paper propose a new scoring function for knowledge base embedding.
The scoring function called TransGaussian is an novel take on (or a generalization of) the well-known TransE scoring function.
The proposed function is tested on two tasks knowledge-base completion and question answering.

----------

OVERALL JUDGMENT
While I think this proposed work is very interesting and it is an idea worth to explore further, the presentation and the experimental section of the paper have some problems.
Regarding the presentation, as far as I understand this is not an attention model as intended standardly in the literature.
Plus, it has hardly anything to share with memory networks/neural Turing machines, the parallel that the authors try to make is not very convincing.
Regarding the experimental section, for a fair comparison the authors should test their model on standard benchmarks, reporting state-of-the-art models.
Finally, the paper lack of discussion of results and insights on the behavior of the proposed model.


----------

DETAILED COMMENTS


In section 2.2 when the authors calculate \mu_{context} do not they loose the order of relations? And if it is so, does it make any sense?

[Official Review · AnonReviewer1 · rating 4 · confidence 4 · 19 Dec 2016]

This paper presents extensions to previous work using embeddings for modeling Knowledge Bases and performing Q&A on them, centered around the use of multivariate gaussian likelihood instead of inner products to score attention. This is supposed to allow more control on the attention by dealing with its spread.

This is a dense paper centered around a quite complicated model. With the supplementary material, this makes a 16p paper. It might be clearer to make 2 separate papers: one on KB completion and another one on Q&A.

I like the idea of controlling the spread of the attention. This makes sense. However, I do not feel that this paper is convincing enough to justify its use compared to usual inner products.

For several reasons:
- These should be more ablation experiments to separate the different pieces of the model and study their influence separately. The only interesting point in that sense is Table 8 in Appendix B. We need more of this. 
- In particular, a canonical experiments comparing Gaussian interaction vs inner product would be very useful. 
- Experiments on existing benchmarks (for KB completion, or QA) would help. I agree with the authors that it is difficult to find the perfect benchmark, so it is a good idea to propose a new one (WorldCup2014). But this should come in addition to experiments on existing data.
- Table 11 of Appendix C (page 16) that compares TransE and TransGaussian for the task of link prediction on WordNet can be seen as fixing the two points above (simple setting on existing benchmark). Unfortunately, TransGaussian does not perform well compared to simpler TransE. This, along with the poor results of TransGaussian (SINGLE) of Table 2, indicate that training TransGaussian seems pretty complex, and hence question the actual validity of this architecture.

[Official Review · AnonReviewer2 · rating 5 · confidence 4 · 21 Dec 2016]

The contribution of this paper can be summarized as:

1, A TransGaussian model (in a similar idea of TransE) which models the subject / object embeddings in a parameterization of Gaussian distribution.  The model can be naturally adapted to path queries like the formulation of (Guu et al, 2015).
2. Along with the entity / relation representations trained by TransGaussian, an LSTM + attention model is built on natural language questions, aiming at learning a distribution (not normalized though) over relations for question answering.
3. Experiments on a generated WorldCup2014 dataset, focusing on path queries and conjunctive queries.

Overall, I think the Gaussian parameterization exhibits some nice properties, and could be suitable to KB completion and question answering. However, some details and the main experimental results are not convincing enough to me.  The paper writing also needs to be improved. More comments below:

[Major comments]

- My main concern is that that evaluation results are NOT strong. Either knowledge base completion or KB-based question answering, there are many existing and competitive benchmarks (e.g., FB15k / WebQuestions). Experimenting with such a tiny WordCup2014 dataset is not convincing.  Moreover, the questions are just generated by a few templates, which is far from NL questions. I am not even not sure why we need to apply an LSTM in such scenario. The paper would be much stronger if you can demonstrate its effectiveness on the above benchmarks. 

- Conjunctive queries:  the current model assumes that all the detected entities in the question could be aligned to one or more relations and we can take conjunctions in the end. This assumption might be not always correct, so it is more necessary to justify this on real QA datasets.

- The model is named as  “Gaussian attention” and I kind of think it is not very closely related to well-known attention mechanism, but more related to KB embedding literature.

[Minor comments]
- I find Figure 2 a bit confusing. The first row of orange blocks denote KB relations, and the second row of those denote every single word of the NL question. Maybe make it clearer?

- Besides “entity recognition”, usually we still need an “entity linker” component which links the text mention to the KB entity.

[Final Decision · Program Chairs · 06 Feb 2017]
**ICLR committee final decision**

Three knowledgable reviewers recommend rejection. While they agree that the paper has interesting aspects, they suggest a more convincing evaluation. The authors did not address some of the reviewer's concerns. The AC strongly encourages the authors to improve their paper and resubmit it to a future conference.